# Isolation of a New Polysaccharide from Dandelion Leaves and Evaluation of Its Antioxidant, Antibacterial, and Anticancer Activities

**DOI:** 10.3390/molecules27217641

**Published:** 2022-11-07

**Authors:** Mo Li, Henan Zhang, Xinyu Hu, Yumeng Liu, Yanfeng Liu, Meijun Song, Rina Wu, Junrui Wu

**Affiliations:** 1College of Food Science, Shenyang Agricultural University, Shenyang 110866, China; 2College of Criminal Science and Technology, Criminal Investigation Police University of China, Shenyang 110035, China; 3Liaoning Engineering Research Center of Food Fermentation Technology, Shenyang 110866, China; 4Shenyang Key Laboratory of Microbial Fermentation Technology Innovation, Shenyang 110866, China

**Keywords:** dandelion leaves, polysaccharides, structural characterization, biological properties, anticancer activity

## Abstract

Dandelion, in China, has a long history as a medicinal and edible plant, and possesses high nutritional and medical value. The present study aimed to isolate a new polysaccharide (DLP-3) from dandelion leaves and to evaluate its antioxidant, antibacterial, and anticancer activities. The structure of DLP-3 was analyzed using HPLC, FT-IR, SEM, GC-MS, and NMR spectroscopy. DLP-3 mainly consisted of Man, Rha, GlcA, Glc, Gal, and Ara with molar ratios of 2.32, 0.87, 1.21, 3.84, 1.00, and 1.05, respectively, with a molecular weight of 43.2 kDa. The main linkages of DLP-3 contained (1→4)-α-d-Glc, (1→4,6)-α-d-Glc, (1→6)-α-d-Gal, (1→2)-α-d-Man, (1→4)-α-d-Man, β-l-Ara-(1→, and α-l-Rha-(1→. DLP-3 exhibited a smooth surface, purely flake-like structure, and a triple helix conformation. Moreover, DLP-3 presented obvious antioxidant and antibacterial activities in a concentration-dependent manner. DLP-3 showed significant anticancer activities by inhibiting tumor cell proliferation. These findings provide a theoretical basis for the application of DLP-3 as a natural functional active substance in functional foods.

## 1. Introduction

Dandelion (*Taraxacum officinale* Weber ex F.H. Wigg.) is widely distributed throughout the Northern Hemisphere [1,2]. In China, dandelion has been one of the most widely used traditional chinese medicines (TCMs) for thousands of years and often found in the Chinese diet as a wild vegetable, and is a type of medicine and food homologous plant. Dandelion contains a variety of functional components, including polysaccharides, tocopherols, riboflavin, and peptides [3,4,5,6,7], and possesses a variety of bioactivities, including antidiabetic, antibacterial, and immunoregulatory activities [8,9,10,11,12]. 

Polysaccharide is a type of macromolecular compound that widely exists in plants, animals, and microorganisms [13,14]. In recent years, natural plant polysaccharides have attracted extensive attention due to their important biological activities, such as promoting the proliferation of probiotics, as well as antitumor, antiviral, bacteriostatic, and immune regulation [15,16,17,18,19]. The bioactivities of polysaccharides are related to their monosaccharide composition, glycosidic bond types, and conformations [20,21,22]. Therefore, studies on the structure–bioactivity relationships of polysaccharides have been very important for its development and application. 

Dandelion polysaccharide (DP) is the main active substance in dandelion. Many studies have confirmed that DP has a variety of biological activities, such as antioxidation, antibacterial, anti-inflammatory, and anticancer activities [23]. DPs extracted from different parts of dandelion have different pharmacological effects. However, most studies have mainly focused on the extraction method, structure, and function of dandelion root polysaccharide [2,24]. To the best of our knowledge, the structure and biological activities (such as antioxidation, antibacterial, and anticancer activities) of purified polysaccharides from dandelion leaves (DLP) have not yet been fully studied, and there are few reports on the structure–bioactivity relationships of DLP [1,6]. Therefore, in this study, dandelion leaves were used as the main raw material, an active polysaccharide (DLP-3) was obtained from dandelion leaves, and its structure properties were characterized by HPLC, FT-IR, SEM, GC-MS, and NMR analyses. Moreover, its antioxidant, antibacterial, and anticancer activities were evaluated. In addition, the structure–bioactivity relationships of DLP-3 were further analyzed. It is expected to provide a reference for the development and application of DLP.

## 2. Results and Discussion

### 2.1. Characterization of DLP-3

#### 2.1.1. Physicochemical Properties and Monosaccharide Composition of DLP-3

DLP-3 was isolated via DEAE-52 and Sephadex G-100 and the elution profiles are shown in Figure 1B,C. The yield of DLP-3 was 4.10 ± 0.29% and the total carbohydrate, protein, and uronic acid contents of DLP-3 were 86.22 ± 4.73%, 2.46 ± 0.88%, and 4.79 ± 0.02%, respectively. The Mw of DLP-3 was evaluated using HPGPC (Figure 2A). Based on the calibration curve (y = −0.2959x + 10.722, R^2^ = 0.9926), the Mw of DLP-3 was determined to be 43.2 kDa. Guo et al. reported a water-soluble polysaccharide PD1-1 from dandelion with a small Mw of 2.6 kDa [25]. Cai et al. reported two polysaccharides (DRP-2b and DRP-3a) from dandelion with Mws of 31.8 kDa and 6.72 kDa, respectively [2]. This result indicated that the Mw of DLP-3 differed from currently reported polysaccharides from dandelion, which might endow DLP-3 some biological activities.

The monosaccharide composition analysis of DLP-3 was evaluated using HPLC. As shown in Figure 2B, DLP-3 consisted of Man, Rha, GlcA, Glc, Gal, and Ara, with molar ratios of 2.32, 0.87, 1.21, 3.84, 1.00, and 1.05, respectively. In addition, the data indicated that DLP-3 contained a large proportion of Man and Glc.

#### 2.1.2. SEM Analysis of DLP-3

The microstructure of DLP-3 was determined using SEM at magnifications of 5000× and 10,000×, which are shown in Figure 3A,B. DLP-3 exhibited a smooth surface and a purely flake-like structure. The apparent structure of polysaccharides can determine their physical and chemical properties [26]. The smooth flake-like structure of DLP-3 may be related to its antioxidant, antibacterial, and anticancer activities.

#### 2.1.3. FT-IR Analysis

The FT-IR spectra of DLP-3 are shown in Figure 3C. The characteristic absorption peaks of DLP-3 can be observed at 3408.75 cm^−1^ and 2928.27 cm^−1^, which are attributed to O-H and C-H groups [27]. The absorption bands of DLP-3 at 1623.49 cm^−1^ and 1410.85 cm^−1^ indicate that there may be -COO- stretching vibration [28]. The absorption peak observed at 1144.26 cm^−1^ may be due to C-O stretching, indicating that a pyran ring structure may exist in DLP-3 [29]. The characteristic absorption at 878.07 cm^−1^ indicates that the main glycosyl unit of DLP-3 is in the β configuration [30]. The absorption peak at 784.57 cm^−1^ indicates that DLP-3 contains mannose.

#### 2.1.4. Methylation Analysis

The linkage patterns of DLP-3 were analyzed using GC–MS. As shown in Table 1, seven components were detected in DLP-3, namely 2,3,6-Me_3_-Glc*p*, 2,3-Me_2_-Glc*p*, 2,3,4-Me_3_-Gal*p*, 3,4,6-Me_3_-Man*p*, 2,3,6-Me_3_-Man*p*, 2,3,5-Me_3_-Ara*f,* and 2.3.4-Me_3_- Rha*p*. These compounds are in molar ratios (%) of 22.29, 14.74, 12.55, 20.19, 11.27, 15.84, and 3.12. By comparing with the mass spectrum patterns from the literature, it is speculated that the linkages of Glc, Gal, Man, Ara, and Rha can be deduced to be (1→4), (1→4,6), (1→6), (1→2), (1→4), and (1→).

#### 2.1.5. NMR Analysis of DLP-3

The structural features of DLP-3 were further identified using an NMR spectral analysis and the entire assignment shifts of the ^1^H and ^13^C for DLP-3 were identified with reference to the previous literature and are illustrated in Appendix A and Table 2 [19,31,32,33,34,35,36]. The anomeric proton signals (5.40, 4.89, 5.10, 5.13, 5,39, 5.20, and 5.39 ppm) and the anomeric carbon signals (98.74, 99.25, 97.67, 100.27, 99.78, 92.22, and 109.26 ppm) correspond to H-1 and C-1 of (1→4)-α-d-Glc, (1→4,6)-α-d-Glc, (1→6)-α-d-Gal, (1→2)-α-d-Man, (1→4)-α-d- Man, β-l-Ara-(1→, and α-l-Rha-(1→. The NMR results are consistent with the results of methylation. In addition, the signal at δ 179.14 ppm should be assigned to the carboxylic group of uronic acids [19].

#### 2.1.6. Congo Red Test

As shown in Figure 3D, the λ_max_ of DLP-3 increases first, then decreases with an increasing concentration of NaOH, which indicates that DLP-3 has a triple helix conformation. Polysaccharides with triple helix conformation have shown higher biological activity than those without this conformation [37,38]. In this study, the triple helix structure of DLP-3 may be related to its strong biological activities.

### 2.2. In Vitro Antioxidant Activities of DLP-3

#### 2.2.1. Superoxide Radical Scavenging Activity

As shown in Figure 4A, the superoxide radical scavenging activity of DLP-3 increased in a DLP-3 concentration-dependent manner, but was slightly less than Vc. The superoxide radical scavenging effect of DLP-3 was 56.49% at 1.0 mg/mL.

#### 2.2.2. ABTS Radical Scavenging Activity

As shown in Figure 4B, DLP-3 exhibited strong ABTS radical scavenging activity in a concentration-dependent manner (0.2–1 mg/mL). The scavenging activity of DLP-3 was 99.12%, which was close to Vc (99.46%) at 1.0 mg/mL.

#### 2.2.3. Hydroxyl Radical Scavenging Assay

As shown in Figure 4C, DLP-3 exhibited strong scavenging activity that increased with an increase in concentration. The scavenging activity of DLP-3 was as high as 88.35% at 1.0 mg/mL.

The antioxidant activity of polysaccharide was speculated to be related to its large amount of hydroxyl groups, which could donate electrons to reduce the radical to a more stable form, or react with the free radicals to terminate the radical chain reaction and inhibit the generation of free radicals by reducing metal ions [34]. The antioxidant activities of DLP-3 were determined by the scavenging activity of the superoxide radicals, ABTS radicals, and hydroxyl radicals. The content of uronic acid, glycosidic linkages, microstructure, monosaccharide composition, and Mw of polysaccharides determine their antioxidant activity [39,40]. Acidic polysaccharides with high uronic acid content have shown stronger free radical scavenging effect and higher antioxidant activity than neutral polysaccharides without uronic acid [41]. Many researchers have reported that low Mw polysaccharides showed stronger antioxidant activities than those with high Mw [31,42]. In addition, a triple helix stereo-configuration and a high level of Glc and Gal have exhibited strong antioxidant activities [43,44]. Moreover, the polysaccharides with a backbone composed of →4)-α-d-Manp-(1→ and →2)-α-d-Manp-(1→ might show good antioxidant activity [44]. Here, the low Mw; the triple helix stereo-configuration; the glycosidic bonds; and the high uronic acid, Glc, and Gal contents of DLP-3 might be the main reasons for antioxidant activities of DLP-3.

### 2.3. Antibacterial Activity of DLP-3

The antibacterial activity of DLP-3 was evaluated against two Gram-negative and two Gram-positive bacteria. As shown in Figure 5, DLP-3 showed a certain inhibitory effect on all strains, and the antibacterial activity increased with an increase in concentration (0.4–2.0 mg/mL). At the DLP-3 concentration of 2.0 mg/mL, the inhibitory diameters against *E. coli*, *Salmonella*, *S. aureus*, and *L.monocytogenes* were 9.4 ± 0.2 mm, 11.6 ± 0.6 mm, 11.8 ± 0.7 mm, and 15.4 ± 0.5 mm, respectively. As compared with previous reports, the antibacterial activity of LDP-3 against *S. aureus* was significantly stronger than that of the polysaccharides extracted from olive leaves (OLP) [45] and the inhibitory activity of LDP-3 against *S. aureus* and *L.monocytogenes* was stronger than that of the polysaccharides isolated from *Periploca laevigata* root Barks (PLP1) [46]. However, DLP-3 showed lower inhibitory activity against *E. coli* than polysaccharides from spent mushroom substrates (PL2) [47] and PLP1 [46]. The data showed that *S.aureus* and *L. monocytogenes*, two Gram-postive bacteria, were the most sensitive to DLP-3. The results of this study are consistent with those of Hashemifesharaki et al. (2020) [48] who reported that marshmallow root polysaccharide showed stronger inhibitory effect on Gram-positive bacteria. Polysaccharides may destroy the inner membrane of cells, increase the permeability of cell membrane, and release electrolytes, resulting in bacterial cell death [48,49]. The inhibitory effects of DLP-3 on Gram-positive bacteria (*S.aureus* and *L.monocytogenes*) were significantly stronger than those on Gram-negative bacteria (*E. coli* and *salmonella*). This result may be due to the presence of lipopolysaccharide in the cell membrane structure of Gram-negative bacteria, which is more complete than Gram-positive bacteria [50].

At the same time, the MIC of DLP-3 against *E. coli*, *salmonella*, *S. aureus,* and *L.monocytogenes* was also investigated to evaluate the antibacterial activity of DLP-3. The results presented that the MICs of DLP-3 against *E. coli*, *salmonella*, *S. aureus,* and *L.monocytogenes* were 1.6, 1.0, 0.2, and 0.6 mg/mL, respectively. All of these suggest that DLP-3 could be considered to be a potential alternative antibacterial agent to currently used antibiotics. Hajji et al. (2018) [51] reported that the MICs of polysaccharides isolated from *Periploca laevigata* root barks (PLP1) against *E. coli*, *S. aureus,* and *L. monocytogenes* were 0.5 mg/mL, 0.3 mg/mL, and 0.8 mg/mL, respectively. Kungel et al. (2018) [46] reported that the MICs of polysaccharides isolated from yerba mate (Ilex paraguariensis) (YM polysaccharide) against *E. coli* and *S. aureus* were 1.6 mg/mL and 0.6 mg/mL, respectively. In this study, DLP-3 showed lower MICs against *S. aureus* and *L. monocytogenes* than PLP1 and lower MICs against *E. coli* and *S. aureus* than YM. Polysaccharides may become a barrier to inhibit the growth of bacterial cells by preventing the intake of nutrients [52]. In addition, high concentrations of polysaccharides can also increase cell permeability and structural damage by destroying the inner membrane, releasing electrolytes and other cellular components, and leading to bacterial cell death [48].

### 2.4. Anticancer Activity of DLP-3

The anticancer effect of DLP-3 against the HepG2 is illustrated in Figure 6. DLP-3 could inhibit the proliferation of HepG2 in a concentration-dependent manner. The results of the MTT assay indicated that the proliferation of HepG2 cells treated with DLP-3 was significantly inhibited (Figure 6A). The MTT results indicated that HepG2 cells were more susceptible to DLP-3 at 72 h.

In the colony formation assay, HepG2 cells treated with DLP-3 (0.2, 0.4, and 0.8 mg/mL) formed fewer colonies than the untreated control group, a phenomenon that was concentration dependent (Figure 6B,C). The results indicated DLP-3 could effectively slow down the clonogenic of HepG2.

The physicochemical properties, structural configuration, and microstructure of polysaccharides determine their anticancer activity [53,54]. The polysaccharide with lower Mw and triple helical chains exhibited stronger binding to receptors on the surface of immune cells due to the occurrence of cross-link receptors, finally leading to higher anticancer activity [55,56]. In addition, the polysaccharide with →4)-α-d-Glcp-(1→ structure has high levels of anticancer and immunoregulatory activities [34]. DLP-3 had high anticancer activity, which might be due to its low molecular weight, triple helix configuration, and glycosidic bonds. Previous studies have shown that polysaccharides can inhibit the proliferation and induce apoptosis of HCC cells by downregulating the activity of the PI3K/Akt/mTOR pathway and upregulating the ratio of Bax/Bcl-2 and activating caspase-3 [3,57].

## 3. Materials and Methods

### 3.1. Materials

The fresh dandelion (variety: *Taraxacum antungense Kitag*.) leaves were purchased from a local market (Shenyang, Liaoning, China) and authenticated by Professor Lin Shi (Shenyang Agricultural University). DEAE-52 cellulose and Sephadex G-100 were obtained from Solarbio Co., Ltd. (Beijing, China). 1-Phenyl-3-methyl-5-pyrazolonde (PMP) and 2,2’-azino-bis-(3-ethylbenzthiazoline-6- sulphonic acid) (ABTS) were purchased from Sigma-Aldrich Chemical Co. (St. Louis, MO, USA). Trypsin EDTA and streptomycin were purchased from Gibco (Grand Island, NY, USA). All other chemicals were of reagent grade.

### 3.2. Extraction and Purification of DLP

The fresh dandelion leaves were washed and dried at 50 °C for 48 h. Then, the dried dandelion leaves were crushed into powders and selected with a 100-mesh sieve. The powders were treated with petroleum to remove fat and extracted with 85% ethanol for 24 h to remove pigments and small organic compounds. After suction filtration, the filter residue was dried and extracted with distilled water (1:20, g/mL) at 80 °C for 2 h. After centrifuging (4000 rpm, 15 min) the extraction solution, the supernatant was concentrated to a proper volume by decompression at 70 °C in a rotary evaporator, and then precipitated with ethanol (1:4, *v/v*) for 12 h, and centrifuged again (4000 rpm, 10 min). Then, the resulting supernatant was deproteinized with the Sevag reagent and precipitated again with ethanol, collected by centrifugation (4000 rpm, 10 min). The precipitates were dialyzed against water for 48 h and lyophilized as crude DLPs. 

The crude DLPs were dissolved and added to a DEAE-52 cellulose column (2.6 cm × 40 cm), and eluted stepwise with NaCl solutions (0–0.5 M). The sugar elute was collected with tubes (5 min/tube) and the absorbance was measured by monitor through the phenol sulfuric acid method. The absorption peak curve was drawn, and then the major fractions were concentrated, dialyzed (cut-off Mw 8000–14,000 Da), and lyophilized. The dried polysaccharides were dissolved and purified with a Sephadex G-100 gel filtration column (1.6 × 30 cm). The top fractions of elution peak (5 min/tube) were collected and lyophilized to obtain the purified DLP. The overall procedure used herein to purify the DLP is schematically illustrated in Figure 1A. The extraction yield of DLP was calculated by Formula (1):(1)$$\mathrm{Extraction}\;\mathrm{yield}\;(\%)={(w}_{1}{/w}_{0})\times 100\%$$
where w_1_ is the weight of DLP and w_0_ is the weight of dry dandelion leaf powder.

DLP-3 with high antioxidant and antibacterial activities was obtained by comparing the antioxidant and antibacterial activities of purified single component polysaccharides (data not provided). Here, we focus on the research of DLP-3.

### 3.3. Characterization of DLP-3

The contents of total carbohydrate, uronic acid, and protein in DLP-3 were measured by using a phenol-sulphuric acid assay [58], m-hydroxydiphenyl assay [59], and coomassie brilliant blue assay [60], respectively. The Mw and microstructure of DLP-3 was analyzed using HPGPC and SEM. The monosaccharide composition and the methylation analysis of DLP-3 were analyzed using HPLC as reported in previous studies [14,20]. The specific experimental methods were presented in S1 and S2. The infrared spectrum and the ^1^H NMR spectra and ^13^C NMR spectra of DLP-3 were determined using FT-IR (4000–400 cm^−1^) and a Bruker AV-400 NMR spectrometer. The CR test of DLP-3 was performed using the method of Nie et al. (2018) [61]. The DLP-3 solution (2.0 mL, 2.5 mg/mL) and CR solution (2.0 mL, 80 μmol/L) were mixed and NaOH solution (1 mol/L) was added to obtain the concentration of NaOH within 0–0.5 M. Then, the λmax was recorded by UV-Vis spectra.

### 3.4. Antioxidant Activity

#### 3.4.1. Superoxide Radical Scavenging Assay

The superoxide radical scavenging assay was determined using the method of Chen et al. (2019) [31] with a minor modification. The specific experimental methods are presented in Appendix A.

#### 3.4.2. ABTS Radical Scavenging Assay

The ABTS radical scavenging assay was determined using the method of He et al. (2018) [62]. The details are provided in Appendix A.

#### 3.4.3. Hydroxyl Radical Scavenging Assay

The hydroxyl radical scavenging assay was measured using the method of Chen and Huang (2019) [63], with a minor modification. The details are provided in Appendix A.

### 3.5. Antibacterial Activity of DLP-3

#### 3.5.1. Agar Diffusion Assay

Inhibitory zones of DLP-3 against Escherichia coli (*E. coli*, CVCC1491), Staphylococcus aureus (*S. aureus*, CVCC1884), Listeria monocytogenes (*L. monocytogenes*, CVCC1597), and *Salmonella* (CVCC2139) were evaluated by the method of Wang et al. (2018) [64]. Briefly, the test organism suspensions (100 μL,10^6^ CFU/mL) were inoculated on the surface of solid medium. The filter paper disks (diameter of 6 mm and thickness of 1 mm) containing DLP-3 (0.4–2.0 mg/mL, 30 μL) were placed in the plates and incubated at 37 °C for 24 h. Sterile water was used as a blank control. The inhibitory zones (diameter expressed in millimeters) of DLP-3 were measured using a vernier caliper.

#### 3.5.2. Minimum Inhibitory Concentration (MIC) Determination

The MIC of DLP-3 was evaluated using the method of Liu et al. (2018) [65]. Briefly, the DLP-3 samples (100 μL) with different concentrations (0.1, 0.2, 0.4, 0.6, 0.8, 1.0, 1.2, 1.4, 1.6, 1.8, and 2.0 mg/mL) and the bacterial suspensions (100 μL, 10^6^ CFU/mL) were added into a 96-well microplate. The absorbance value of the culture medium was measured after incubated at 37 °C for 24 h. The MIC of DLP-3 was determined.

### 3.6. Anticancer Properties

#### 3.6.1. Cell Lines and Cell Culture

The HepG2 cells were obtained from the Stem Cell Bank, Chinese Academy of Sciences and cultured in Dulbecco’s modified Eagle medium (DMEM) with 10% fetal bovine serum (FBS) and 1% penicillin in a incubator at 37 °C with 5% CO_2_.

#### 3.6.2. Cell Proliferation Assay and Colony Formation Assay

The MTT cell proliferation assay and the colony formation assay were evaluated following a previously reported method [48,66]. The specific experimental methods are presented in Appendix A.

### 3.7. Statistical Analysis

All experiments were conducted three times. The data were analyzed using the SPSS 19.0 statistical software. The results are reported as means ± SD. The difference was *p* < 0.05.

## 4. Conclusions

In the present study, we demonstrated a novel polysaccharide (DLP-3) that possessed good biological activity which was successfully isolated from dandelion leaves. DLP-3 had an average MW of 43.2 kDa and was composed of Man, Rha, GlcA, Glc, Gal, and Ara. DLP-3 with a triple helix structure exhibited high antioxidant, antibacterial, and anticancer activities in vitro. These results indicate that the biological activity of DLP-3 is greatly determined by its monosaccharide composition, MW, and triple helix conformation. In conclusion, DLP-3 can be used as an active substance in the development of functional food and pharmaceutical products. However, the molecular mechanisms of antioxidant, antibacterial, and anticancer effects of DLP-3 should be investigated further.

## Figures and Tables

**Figure 1 molecules-27-07641-f001:**
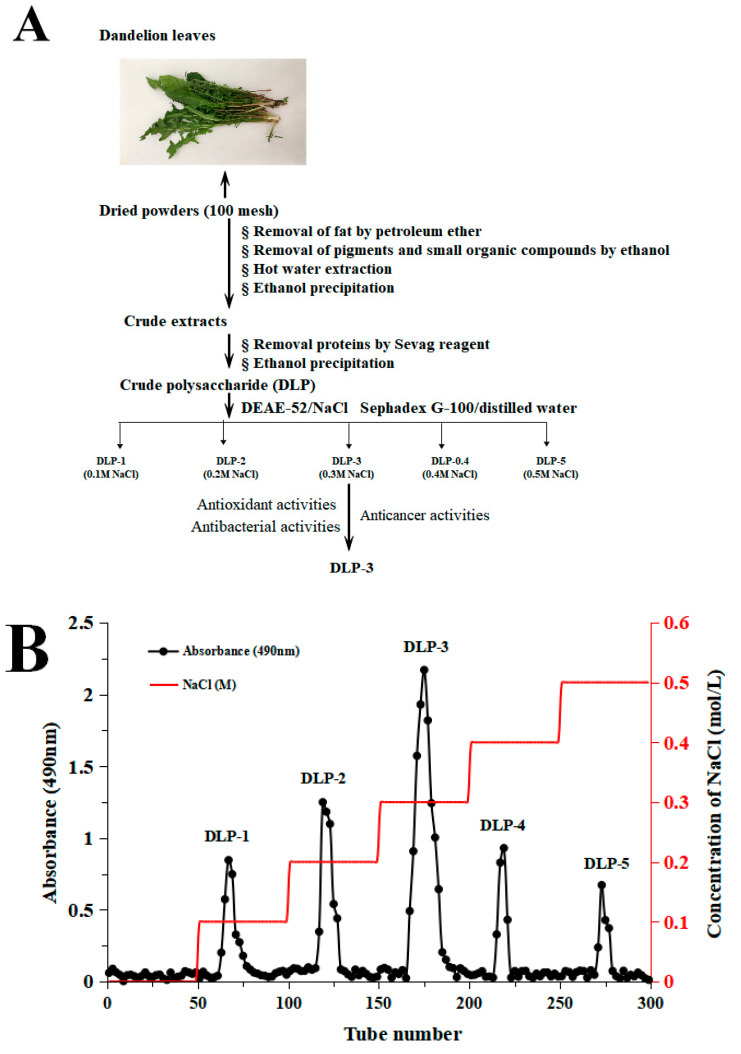
(**A**) Flow chart of extraction and purification of DLP-3; (**B**) chromatography of DLP-3 using a DEAE-52 cellulose column; (**C**) Sephadex G-100 chromatography.

**Figure 2 molecules-27-07641-f002:**
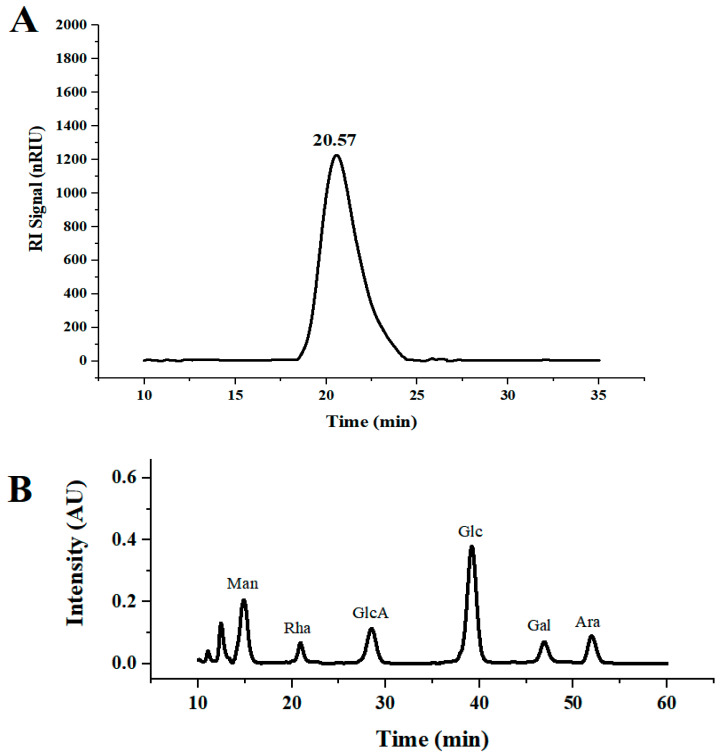
(**A**) HPGPC chromatograms of DLP-3; (**B**) monosaccharide composition of DLP-3.

**Figure 3 molecules-27-07641-f003:**
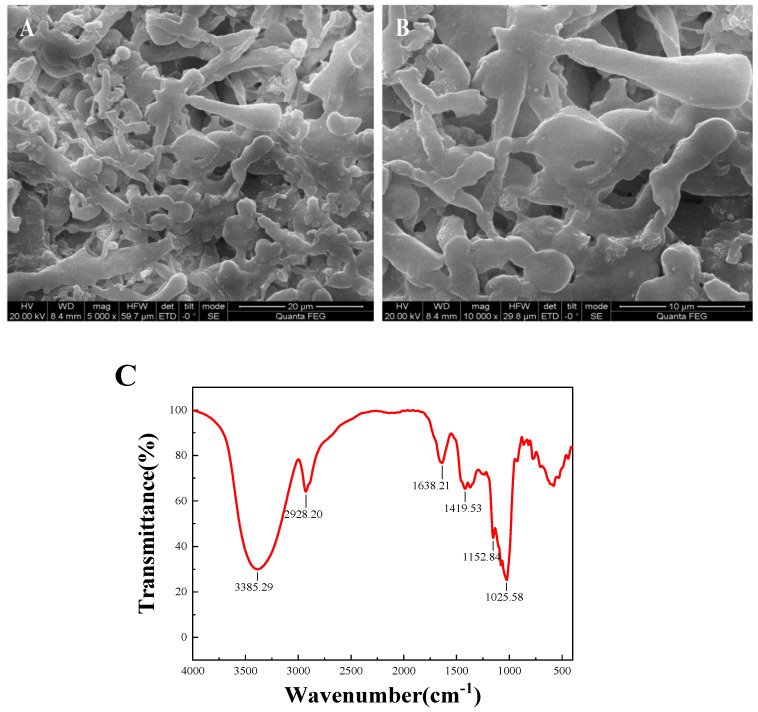
(**A**,**B**) SEM (5000× and 10,000×) of DLP-3; (**C**) FT-IR spectra of DLP-3 in the range of 4000–400 cm^−1^; (**D**) changes in absorption wavelength maximum of a mixture of Congo red and DLP-3 at various concentrations of NaOH.

**Figure 4 molecules-27-07641-f004:**
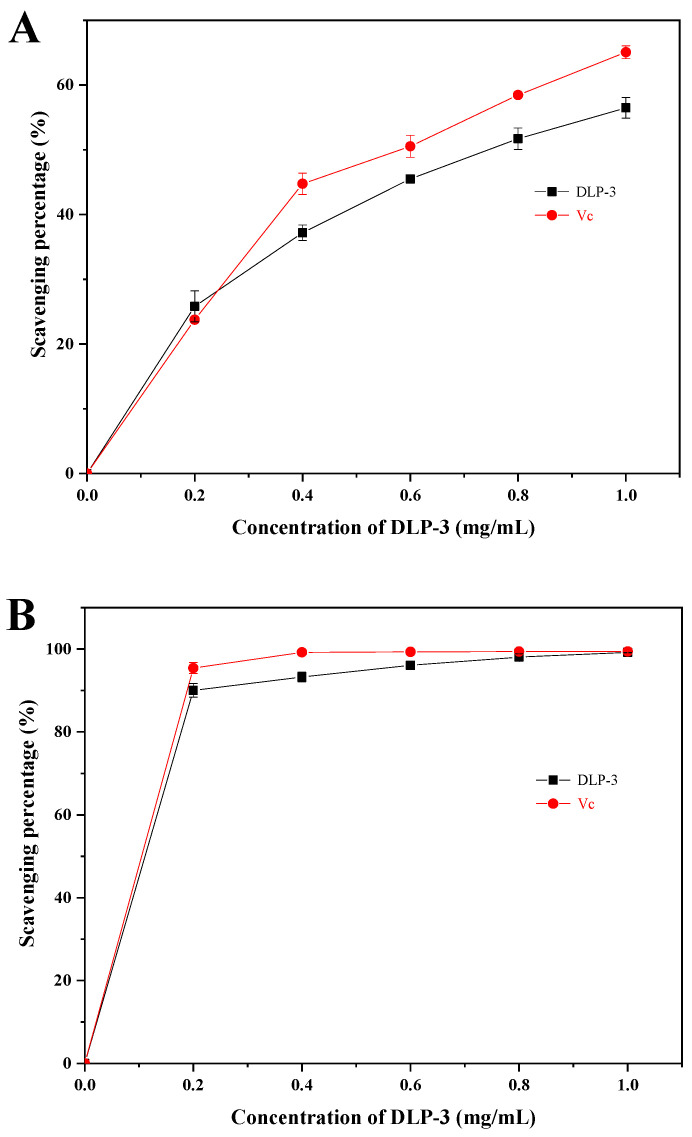
The antioxidant activities of DLP-3: (**A**) Superoxide radicals scavenging activity; (**B**) ABTS radical scavenging activity; (**C**) hydroxyl radical scavenging activity.

**Figure 5 molecules-27-07641-f005:**
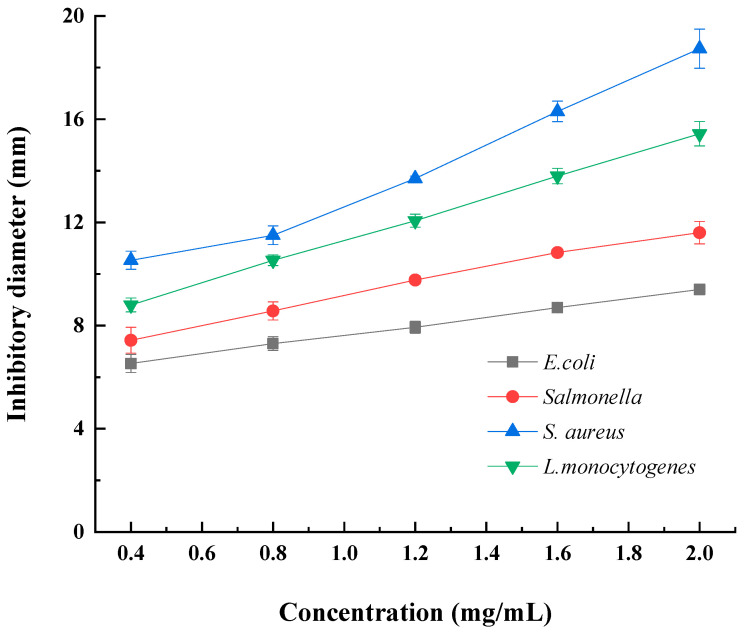
The agar disc diffusion test was applied to assess the antimicrobial activity of DLP-3.

**Figure 6 molecules-27-07641-f006:**
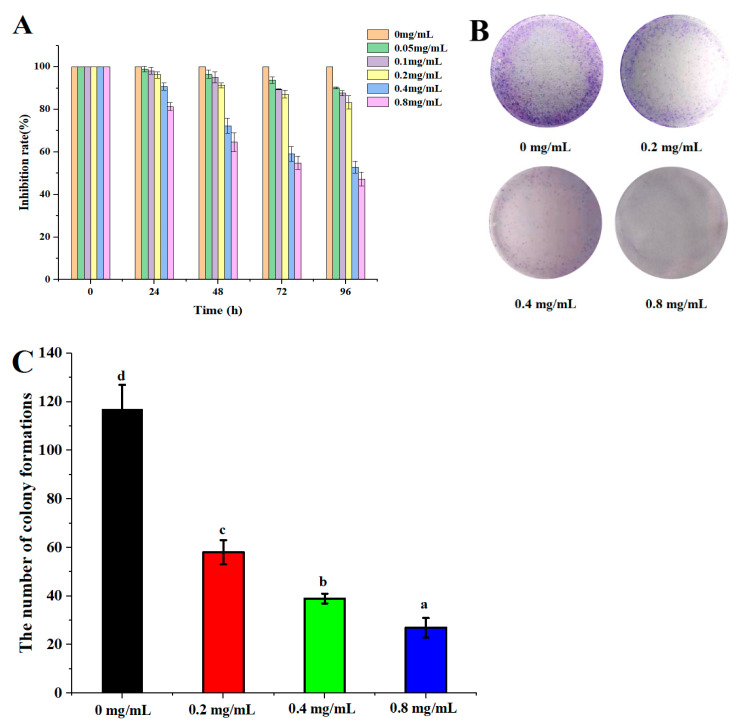
DLP-3 inhibits HepG2 cells proliferation in vitro. HepG2 cells were treated with different concentrations (0, 0.2, 0.4, and 0.8 mg/L) of DLP-3 for 0, 24, 48, 72, and 96 h: (**A**) The effect of DLP-3 on cell proliferation was evaluated using an MTT assay; (**B**,**C**) the clonogenicity of the indicated cells were detected after treatment with different concentrations of DLP-3. Data are presented as the mean ± SD, n = 3. Different letters above the bars indicate significant difference.

**Table 1 molecules-27-07641-t001:** Glycosyl linkage analysis of DLP-3.

Numbers	Characteristic Fragments (*m*/*z*)	Methylated Sugar	Molar Ratio (%)	Linkage Type
1	43, 45, 57, 85, 87, 99, 101, 113, 117, 161, 173, 233	2,3,6-Me_3_-Glc*p*	22.29	→4)-Glc*p*-(1→
2	43, 85, 102, 118, 127, 162, 201, 261	2,3-Me_2_-Glc*p*	14.74	→4,6)-Glc*p*-(1→
3	43, 45, 71, 87, 99, 101, 117, 129, 161, 189, 233	2,3,4-Me_3_-Gal*p*	12.55	→6)-Gal*p*-(1→
4	43, 45, 71, 87, 99, 101, 113, 129, 161, 173, 205	3,4,6-Me_3_-Man*p*	20.19	→2)-Man*p*-(1→
5	43, 45,59, 71, 87, 102, 113, 118, 129, 162, 173, 233	2,3,6-Me_3_-Man*p*	11.27	→4)-Man*p*-(1→
6	59, 71, 87, 102, 118, 129, 145, 161	2,3,5-Me_3_-Ara*f*	15.84	Ara*f*-(1→
7	57,72,89,102,113,118,162	2,3,4-Me_3_-Rha*p*	3.12	Rha*p*-(1→

**Table 2 molecules-27-07641-t002:** ^13^C and ^1^ H NMR chemical shifts (ppm, δ) for DLP-3.

Sugar Residue	C1/H1	C2/H2	C3/H3	C4/H4	C5/H5	C6/H6
→4)-α-d-Glcp-(1→	5.40	3.60	3.72	3.60	3.72	4.01
	98.74	67.59	71.16	72.78	69.27	67.59
→4,6)-α-d-Glcp-(1→	4.89	3.54	3.88	3.54	4.01	3.80
	99.25	69.27	73.46	74.38	70.88	64.10
→6)-α-d-Galp-(1→	5.10	3.66	4.03	3.83	4.28	4.03
	97.67	72.78	76.48	70.48	76.48	68.58
→2)-α-d-Manp-(1→	5.13	4.28	-	-	-	3.74
	100.27	74.02	-	-	-	61.28
→4)-α-d-Manp-(1→	5.39	3.88	3.43	3.72	3.52	3.72
	99.78	63.24	75.20	80.71	74.38	69.85
β-l-Araf-(1→	5.20	4.01	4.12	3.80	3.70	-
	92.22	81.37	76.48	77.96	63.24	-
α-l-Rhap-(1→	5.39	3.83	3.66	3.45	4.01	1.27
	109.26	76.48	67.59	71.69	69.27	15.41

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
