# Peer review of "Isolation of a New Polysaccharide from Dandelion Leaves and Evaluation of Its Antioxidant, Antibacterial, and Anticancer Activities"

_molecules, 2022, doi:10.3390/molecules27217641_

Round 1

Reviewer 1 Report

The manuscript reports a good level of data.

The minor check is required for the English language and style.

lines 14, 22, 26 - The italic type should not be used for writing English name of this species  'Dandelion'.

I suppose that 'Polysaccharide' (line 33) should be written in plural 'Polysaccharides'

If dry dandelion leaves bought in the market were used in the study, why are fresh dandelion leaves shown in the photo (Fig. 1)?

line 20 'Moreover, DLP-3 showed a significantly anticancer activity' - This sentence should be reconstructed grammatically.

 The italic type should be used everywhere for writing Latin names of genus and species as well as for 'in vitro' (lines 344, 517, etc).

Why weren't used modern references published in 2022 and only two for 2021 year? Such publications can be easily found in PubMed or Scopus databases. These additions would significantly strengthen the novelty of the chosen topic.

Author Response

Manuscript ID (Molecules-1978805)

Title: Purification, characterization of an active polysaccharide from Dandelion leaves and evaluation of antioxidant, antibacterial and anticancer activity

Correspondence Author: Rina Wu

Dear editor and Reviewers:

I am very grateful to your comments for the manuscript. According with your comments, we amended the relevant part in manuscript (the revised parts are marked in red in the manuscript). In addition, we have had the manuscript polished with a professional assistance in writing and revised the format according to the requirements of the journal. Some of your questions were answered below.

Reviewer 1:

Comment 1:The minor check is required for the English language and style.

Response: Thank you for your helpful advice, we have had the manuscript polished with a professional assistance in writing and revised the format according to the requirements of the journal.

Comment 2: lines 14, 22, 26 - The italic type should not be used for writing English name of this species  'Dandelion'.

Response: Thanks for your suggestion.We have modified it according to your suggestion.
Comment 3: I suppose that 'Polysaccharide' (line 33) should be written in plural 'Polysaccharides'.

Response: Thanks for your suggestion.We have modified it according to your suggestion.
Comment 4: 2.1 If dry dandelion leaves bought in the market were used in the study, why are fresh dandelion leaves shown in the photo (Fig. 1)?

Response: Thank you very much for your question. The raw material we use is fresh Dandelion leaves purchased in the market, which are washed and dried before polysaccharide extraction. This part is described in Method 2.2, and the inaccurate information is supplemented. Thank you again for your careful review.

Comment 5: line 20 'Moreover, DLP-3 showed a significantly anticancer activity' - This sentence should be reconstructed grammatically.

Response: Thanks for your suggestion.We have modified it according to your suggestion.
Comment 6: The italic type should be used everywhere for writing Latin names of genus and species as well as for 'in vitro' (lines 344, 517, etc).

Response: Thanks for your suggestion.We have modified it according to your suggestion.
Comment 7: Why weren't used modern references published in 2022 and only two for 2021 year? Such publications can be easily found in PubMed or Scopus databases. These additions would significantly strengthen the novelty of the chosen topic.

Response: Thanks for your suggestion. We have supplemented the latest references.

We sincerely hope this manuscript will be finally acceptable to be published on Molecules. Thank you very much for all your help and looking forward to hearing from you soon.

With regards,

Rina Wu

College of Food Science

Shenyang Agricultural University

Shenyang 110866, China

Tel: +86+024+86487161

Fax: +86+024+86487161

wrn6956@163.com

Reviewer 2 Report

In this study authors describe purification and characterization of an active polysaccharide from Dandelion leaves as well as evaluation of antioxidant, antibacterial and anticancer activity.

The paper is well written, but minor revision before publishing (listed in later text) is needed. All technical errors need to be corrected:

- Variables throughout the text are not in italics. Please go through the entire text in detail and correct this.

- In many cases there is no space between the variable and the measuring unit (For example in lines 65, 67, 70, 72, 77, 78, 82, 116, 137, 139, 145, 152…). Please go through the entire manuscript and supplementary in detail and correct this.

- Figures S2 and S3 in supplementary should be S1 and S2

- The concentrations in mg/mL are mass concentrations. I kindly ask the authors to emphasize this in the text and correct it where necessary (figures, tables, supplementary...)

Author Response

Manuscript ID (Molecules-1978805)

Title: Purification, characterization of an active polysaccharide from Dandelion leaves and evaluation of antioxidant, antibacterial and anticancer activity

Correspondence Author: Rina Wu

Dear editor and Reviewers:

I am very grateful to your comments for the manuscript. According with your comments, we amended the relevant part in manuscript (the revised parts are marked in red in the manuscript). In addition, we have had the manuscript polished with a professional assistance in writing and revised the format according to the requirements of the journal. Some of your questions were answered below.

Reviewer 2:

Comment 1:Variables throughout the text are not in italics. Please go through the entire text in detail and correct this.

Response: Thanks for your suggestion.We have modified it according to your suggestion.

Comment 2: In many cases there is no space between the variable and the measuring unit (For example in lines 65, 67, 70, 72, 77, 78, 82, 116, 137, 139, 145, 152…). Please go through the entire manuscript and supplementary in detail and correct this.

Response: Thanks for your suggestion.We have modified it according to your suggestion.
Comment 3: Figures S2 and S3 in supplementary should be S1 and S2.

Response: Thanks for your suggestion.We have modified it according to your suggestion.
Comment 4: The concentrations in mg/mL are mass concentrations. I kindly ask the authors to emphasize this in the text and correct it where necessary (figures, tables, supplementary...)

Response: Thank you for your helpful advice, according to the comments, we have revised these issues. Thank you again.

We sincerely hope this manuscript will be finally acceptable to be published on Molecules. Thank you very much for all your help and looking forward to hearing from you soon.

With regards,

Rina Wu

College of Food Science

Shenyang Agricultural University

Shenyang 110866, China

Tel: +86+024+86487161

Fax: +86+024+86487161

wrn6956@163.com

Reviewer 3 Report

The manuscript "Purification, characterization of an active polysaccharide from Dandelion leaves and evaluation of antioxidant, antibacterial and anticancer activity" is interesting. After making the following corrections, Molecules Journal may consider this manuscript for publication.

1.    I suggest altering the title to "Isolation of new polysaccharide from Dandelion leaves and evaluation of its antioxidant, antibacterial and anticancer activities", which is more accurate.

2.    I was unable to locate the study's objective in the abstract. Insert it in the abstract following the first sentence. “The present study aimed to isolate a new polysaccharide (DLP-3) from Dandelion leaves and evaluate its antioxidant, antibacterial and anticancer activities. Remove the sentenceIn this study, an active polysaccharide (DLP-3) was first isolated from 13 Dandelion leaves”.

3.    The methods portion was nowhere to be found in the abstract. Before addressing the findings in the abstract, include a brief methodology (Method employed, how the characterization was done, etc.).

4.    Line 36 – Change anti-virus to antiviral

5.    Line 36 – Change bacteristasis to bacteriostatic

6.    I couldn't locate any pertinent information concerning the author's selection to perform antioxidant, antibacterial, and anticancer activities on DLP-3 in the introduction. Give a succinct explanation of the purpose and necessity of these biological activities.

7.    Line 46-48, references must be cited in support of the statement.

8.    r/min to be changed as rpm (throughout the manuscript)

9.    Section 2.3.1 to 2.3.8 – Authors should think about combining the paragraphs under "Characterization of DLP-3" into one. It's not required to describe the specific instruments used for characterisation in one by one with detail.

10.  State each microorganism's full name before it is abbreviated. For example, Escherichia coli (E. coli).

11.  The spacing between each abbreviated organism name should be consistent throughout the manuscript. E. coli (Not E.coli), S. aureus (Not S.aureus) and L. monocytogenes (Not L.monocytogenes).

12.  Figure 1, Change the word antioxidant activities to antioxidant activity. Similar changes are being made to the term "antibacterial activity." Where is Anticancer?

13.  The results and discussion section have excellent writing. However, in the conclusion, instead of summarizing the research that has been done, I suggest offering a critical defence of their results/observations. I also notice a few sentences that are repeated from the abstract. Potential points of view must be addressed in the conclusion. The importance of this research article should be highlighted by the author.

Author Response

Manuscript ID (Molecules-1978805)

Title: Purification, characterization of an active polysaccharide from Dandelion leaves and evaluation of antioxidant, antibacterial and anticancer activity

Correspondence Author: Rina Wu

Dear editor and Reviewers:

I am very grateful to your comments for the manuscript. According with your comments, we amended the relevant part in manuscript (the revised parts are marked in red in the manuscript). In addition, we have had the manuscript polished with a professional assistance in writing and revised the format according to the requirements of the journal. Some of your questions were answered below.

Reviewer 3:

Comment 1: I suggest altering the title to "Isolation of new polysaccharide from Dandelion leaves and evaluation of its antioxidant, antibacterial and anticancer activities", which is more accurate.

Response: Thank you for your helpful advice. We have revised the title according to your suggestion.

Comment 2: I was unable to locate the study's objective in the abstract. Insert it in the abstract following the first sentence. “The present study aimed to isolate a new polysaccharide (DLP-3) from Dandelion leaves and evaluate its antioxidant, antibacterial and anticancer activities. Remove the sentence “In this study, an active polysaccharide (DLP-3) was first isolated from 13 Dandelion leaves”.

Response: Thanks for your suggestion.We have modified it according to your suggestion.
Comment 3: The methods portion was nowhere to be found in the abstract. Before addressing the findings in the abstract, include a brief methodology (Method employed, how the characterization was done, etc.).

Response: Thanks for your suggestion. We have supplemented a brief methodology in the abstract.
Comment 4: Line 36–Change anti-virus to antiviral

Response: Thanks for your suggestion.We have modified it according to your suggestion.

Comment 5: Line 36–Change bacteristasis to bacteriostatic.

Response: Thanks for your suggestion.We have modified it according to your suggestion.

Comment 6: I couldn't locate any pertinent information concerning the author's selection to perform antioxidant, antibacterial, and anticancer activities on DLP-3 in the introduction. Give a succinct explanation of the purpose and necessity of these biological activities.

Response: Thanks for your suggestion.We have revised the introduction section according to your suggestion.
Comment 7: Line 46-48, references must be cited in support of the statement.

Response: Thanks for your suggestion. We have supplemented the references according to your suggestion.

Comment 8: r/min to be changed as rpm (throughout the manuscript).

Response: Thanks for your suggestion.We have modified it according to your suggestion.
Comment 9: Section 2.3.1 to 2.3.8 – Authors should think about combining the paragraphs under "Characterization of DLP-3" into one. It's not required to describe the specific instruments used for characterisation in one by one with detail.

Response: Thanks for your suggestion.We have revised the method section according to your suggestion.
Comment 10: State each microorganism's full name before it is abbreviated. For example, Escherichia coli (E. coli).

Response: Thanks for your suggestion.We have modified it according to your suggestion.
Comment 11: The spacing between each abbreviated organism name should be consistent throughout the manuscript. E. coli (Not E.coli), S. aureus (Not S.aureus) and L. monocytogenes (Not L.monocytogenes).

Response: Thank you for your careful review.We have modified it according to your suggestion.
Comment 12: Figure 1, Change the word antioxidant activities to antioxidant activity. Similar changes are being made to the term "antibacterial activity." Where is Anticancer?

Response: Thank you for your careful review.We have modified it according to your suggestion.

Comment 13: The results and discussion section have excellent writing. However, in the conclusion, instead of summarizing the research that has been done, I suggest offering a critical defence of their results/observations. I also notice a few sentences that are repeated from the abstract. Potential points of view must be addressed in the conclusion. The importance of this research article should be highlighted by the author.
Response: Thank you for your careful review.We have rewritten the conclusion based on your suggestion.

We sincerely hope this manuscript will be finally acceptable to be published on Molecules. Thank you very much for all your help and looking forward to hearing from you soon.

With regards,

Rina Wu

College of Food Science

Shenyang Agricultural University

Shenyang 110866, China

Tel: +86+024+86487161

Fax: +86+024+86487161

wrn6956@163.com

Round 2

Reviewer 3 Report

The author responded to all of my inquiries and revised the manuscript content accordingly. As a result, I suggest that it be taken into consideration for publication in Molecules Journal in its current form.